# Seed Germination and Seedling Growth of Yellow and Purple Passion Fruit Genotypes Cultivated in Ecuador

William Viera [1,2], Takashi Shinohara [1,*], Atsushi Sanada [1], Naoki Terada [1], Lenin Ron [3] and Kaihei Koshio [1]

[1] Faculty of International Agriculture and Food Studies, Tokyo University of Agriculture, Sakura gaoka 1-1-1, Setagaya, Tokyo 156-8502, Japan
[2] Fruit Program, Tumbaco Experimental Farm, Instituto Nacional de Investigaciones Agropecuarias (INIAP), Av. Interoaceánica km 15 and Eloy Alfaro, Tumbaco 170902, Ecuador
[3] Faculty of Veterinary Medicine and Zoothecnics, Universidad Central del Ecuador (UCE), Quito 170521, Ecuador
* Correspondence: t3shinoh@nodai.ac.jp; Tel.: +81-3-5477-2207

**Abstract:** Seed is a fundamental tool to carry out breeding processes and for the propagation of the crops; however, seed propagation generally has low and irregular germination. Passion fruit (*Passiflora*) species are economically important for Ecuador, which is the main exporter of passion fruit concentrate in Latin America. Ecuadorian farmers propagate new plants by seeds to establish new passion fruit orchards or to extend their cultivated area. The objective of this research was to determine the differences in germination and seedling development with the application of priming methods in five genotypes of passion fruit belonging to three different taxa that are of commercial use in Ecuador. The genotypes used were: INIAP 2009 and P10 (*P. edulis* f. *flavicarpa*), Gulupa (*P. edulis* f. *edulis*), and local germplasms POR1 (*P. edulis* f. *flavicarpa*) and PICH1 (*P. maliformis*). The priming methods were: water (control), hydrogen peroxide at 15%, potassium nitrate at 1%, PEG 6000 at −1.2 MPa, and gibberellic acid at 500 ppm. The results showed that there was a genotype-response depending on the priming method. Nevertheless, Polietilenoglicol (PEG 6000) could be considered as a promising method to encourage seed germination and promote seedling growth in the *Passiflora* species. More research regarding the use of this compound has to be carried out in order to determine in depth the physiological processes related to its functions to improve seed germination as well as production of vigorous seedlings.

**Keywords:** aged seed; genotypes; Passifloraceae; PEG 6000; priming; seedling vigor

## 1. Introduction

Plant genetic diversity, especially in crop landraces, is fundamental for the mass selection or hybridization to generate improved cultivars that are able to cope with climatic changes and pests, and to increase agricultural production and sustainability [1]. Consequently, plant seed is an essential tool to carry out these breeding processes.

Seed is the main reproductive organ of most of higher plants and it plays an essential role in the renewal, persistence, spread, and regeneration of plant populations [2]. Seed germination is a physiological characteristic that have great impact on adaptation and survival of plants [3]. It is the first critical step of the plant life cycle [4] and is influenced by environmental factors [3].

The passion fruit (*Passiflora* L.) species belong to the Passifloraceae family. Worldwide there are about 600 known species, and more than 500 species produce fruits for fresh consumption and industrial processing. Most of the species are found in South America (Tropical), in countries such as Brazil, Colombia, Peru, Ecuador, Argentina Bolivia, and Paraguay; however, there are native species from the United States, Central America, Asia, Australia, and China [5]. This fruit crop is commercially and economically important; therefore, some breeding works have been carried out [6].

One of the problems observed in passion fruit is its propagation method, mainly by seeds, which generally shows low and irregular germination [7]; in addition, there is little information about germination of *Passiflora* seeds [8]. Priming can be used to encourage seed germination [9]. It is appreciated for seed bank operators who need improved protocols of ex situ germplasm conservation (crop and native species); however, different priming treatments can be applied depending on the plant species, seed morphology, and physiology, triggering the so-called pre-germinative metabolism [10].

Doijode [11] mentioned that passion fruit germination can last from ten days to three months, showing low germination rate and irregular seedling development. As passion fruit seeds exhibit low germination [12–14], different chemical methods have been tested to improve the germination potential. Some studies have been carried out in the main commercial taxa such as *P. edulis* f. *flavicarpa* and just a few *P. edulis* f. *edulis*; however, there is almost no information in less domesticated species such as *P. maliformis,* which is also grown in less proportion for commercialization and due to its soil disease resistance characteristics [15,16].

Priming methods can be used to improve seed germination in different horticultural species such as passion fruit. Hydrogen peroxide ($H_2O_2$), which is an endogenous reactive oxygen species (ROS), has demonstrated in legume species to stimulate early germination and growth [17], but there is little information about its use in the family Pasifloraceae [18]. Martínez et al. [18] found that this chemical can reach more than 80% of germination in passion fruit. Potassium nitrate ($KNO_3$) enhances seed germination by forming a water potential equilibrium between seeds and the solution by osmosis [19].

Polietilenoglicol with molecular weight of 6000 (PEG 6000) is a natural polymer that is water-soluble and nonionic [20]. This compound is considered a promising method for seed germination, especially in water-restricted environments [21]. It has been reported that this compound positively influenced yellow passion fruit seed germination and greater seedling uniformity [8]. In addition, it has been stated that gibberellic acid ($GA_3$) promotes seed germination [4] and it is being used to promote seed germination in several fruit crops [22] including passion fruit [13].

On the other hand, acid treatments have also been used to break down thick impermeable seed coats that constitute a barrier for seed germination. Mabundza et al. [9] reported that chemical scarification using sulfuric acid reached high seed germination in passion fruit. However, care must be taken with the time of application in the seed to avoid damage to the embryo, thus its management could be complicated for farmers.

In Ecuador, passion fruit producers propagate this fruit crop mainly by seeds. There are several passion fruit genotypes that are grown for their national and international commercialization, which are different in their morphological traits as well as their composition of bioactive compounds and minerals [23,24]. Plant nurseries and farmers do not apply any kind of priming method, neither imbibition into water in most cases, before sowing seeds into grown seedlings to transplant to the field to set new orchards or extend the cultivated area of this fruit crop.

Therefore, considering that *Passiflora* species are a resource for passion fruit breeding programs for different approaches such as food and medical purposes, this research had the objective to determine the differences in germination and seedling development with the application of priming methods in five genotypes of passion fruit belonging to three different taxa that are of commercial use in Ecuador.

## 2. Materials and Methods

### 2.1. Experimental Site

This study was done in the laboratory of Tropical Horticultural Science (imbibition and moisture tests) and in the greenhouse of the Department of International Agricultural Development (germination and seedling development); both belonging to the Tokyo University of Agriculture located at an altitude of 53 masl, latitude of 35°38′33″, and longitude of 139°37′48″. The experiment was carried out from February to April 2022.

## 2.2. Seed Material

Seeds of five passion fruit genotypes (Figure S1) were obtained from passion fruit plants of the ex-situ germplasm collection of the National Institute of Agricultural Research (INIAP). To obtain the fruits, artificial pollination was carried out (manually using a brush) taking the pollen from the flower of a plant other than the pollinated one. The genotypes used were: INIAP 2009 and P10 (*P. edulis* f. *flavicarpa* Deneger), Gulupa (*P. edulis* f. *edulis* Sims), and local germplasms POR1 (*P. edulis* f. *flavicarpa*) and PICH1 (*P. maliformis* L.). *P. edulis* f. *flavicarpa* and *P. maliformis* are considered yellow passion fruit while *P. edulis* f. *edulis* is called purple passion fruit. The yellow passion fruit genotypes (INIAP 2009, P10, POR1, and PICH1) were cultivated in the tropical area of Ecuador. INIAP 2009, P10, and POR1 grew at an altitude of 52 masl, an average temperature of 26 °C, an annual rainfall of 852 mm, and 1385 light hours (Manabí province), while PICH1 grew at an altitude of 74 masl, an average temperature of 25 °C, an annual precipitation of 1200 mm, and 920 light hours (Los Ríos province). On the other hand, Gulupa was grown in the inter-Andean valleys, at an altitude of 2348 masl, an average temperature of 17 °C, an annual rainfall of 892 mm, and 2039 light hours (Pichincha province).

The fruits to obtain the seed were chosen at random. Two fruits were taken from 20 independent plants of each genotype, seed was obtained and then mixed in a seed pool to proceed to randomly obtain the seeds to be evaluated. Seeds were dried for three days at room temperature (25 °C), then placed in plastic bags (Ziploc$^{TM}$), sealed, and conserved for 18 months in a cold room at 6 °C and 45% of relative humidity. These genotypes are different phenotypically and in their pulp's phytochemical composition. Their characteristics (check by codification) are described by Viera et al. [23,24].

## 2.3. Chemical Priming Treatments

Initially, the passion fruit seeds for each genotype were placed in a glass with water Type II to eliminate those that floated (empty seeds), the others were dried and used for the tests. Dry seeds were placed into each chemical priming treatment, thus they were placed in a 100 mL glass container and 50 mL of each chemical was added to cover the seeds, except in the case of PEG 6000, where seeds were placed in a petri dish containing two layers of filter paper and the seeds were half-soaked. The treatments that were selected based on previous studies were: pure water Type II used as control and seed remained in it for 24 h; for 48 h in hydrogen peroxide at 15% [18]; for 24 h in potassium nitrate at 1% [25]; for 36 h in PEG 6000 at −1.2 MPa [8]; and for 10 min in gibberellic acid at 500 ppm [13]. All seeds were placed in an incubator (Sansho, CN-25C, Tokyo, Japan) in dark condition at 25 °C according to the time for each treatment. Then, they were washed with pure water Type II and dried for further sowing.

## 2.4. Seed Sowing and Plant Growth

After the chemical priming solutions, seeds were sown in plastic trays containing granular clay soil (Akadama, Japanese volcanic soil) and manure compost in a proportion of 70 and 30%, respectively. There were three replications of 50 seeds and each one was used for each treatment. The trays were irrigated all days, twice per day, for 10 min, using an aspersion system with a caudal of 0.83 L min$^{-1}$. No fertilization was carried out, but the compost had a composition of 1.3% of nitrogen, 1.9% of phosphorous, and 2.3% of potassium. The greenhouse had heating and ventilating system to hold the temperature at 28 °C ± 3 °C and at a relative humidity of 50 ± 3%. There were 10 h of daylight in February, 11 h in March, and 12 h in April. Seedling growth was kept for 60 days after sowing (DAS).

## 2.5. Seed Variables

### 2.5.1. Imbibition Percentage (%)

Fresh seed weight was recorded at 0, 12, 24, 48, and 72 h. Ten seeds were removed for each chemical priming treatment in each evaluation for recording the data, and it was done in triplicate. They were blotted using paper towel and directly weighed on a semi-micro

analytical balance (A&D Company, GH-252, Tokyo, Japan). This parameter was estimated by the formula [26]:

$$I = \left( \frac{FW - IFW}{IFW} \right) \times 100$$

where, I is imbibition percentage, FW is seed fresh weight, and IFW is seed initial fresh weight.

### 2.5.2. Seed Moisture (%)

After each imbibition period, ten seeds (by triplicate) were placed in an oven (Sanyo, MOV-112F, Osaka, Japan) at 80 °C for 48 h, and following that, seeds were weighed in a precision balance. This parameter was calculated by the following formula [26]:

$$SM = \left( \frac{FW - DW}{FW} \right) \times 100$$

where, SM is seed moisture, FW is seed fresh weight, and DW is seed dry weight.

### 2.5.3. Seed Length, Width, Thickness (mm) and Weight (mg)

Seed length, width, and thickness were measured using a caliper (Mitutoyo, CD-15C, Kanagawa, Japan). Thirteen seeds of each genotype were used for the measurements.

Ten seeds of each genotype were also weighted using a semi-micro analytical balance. This was replicated ten times (A&D Company, GH-252, Tokyo, Japan).

### 2.5.4. Germination Percentage (%)

Percentage was recorded every 24 h until 60 DAS. To be considered as germination, firstly the epicotyl was observed over the soil (seedling emergency) and then the seedling developed (Figure S2); consequently, at 20 DAS, it was considered as seed emergence. The following formula [27] was used:

$$GP = \left( \frac{NG}{TN} \right) \times 100$$

where, GP is germination percentage, NG is the number of germinated seeds, and TN is the total number of seeds.

### 2.5.5. Day of the First Plant Emergence, Gt50 and Germination Rate

The day in which the first plant (epicotyl) emerged for each genotype was recorded.

The number of days needed to reach 50% of germination (Gt50) was estimated based on the germination per day.

Finally, germination rate was estimated using the following formula [27]:

$$GR = \Sigma \left( \frac{NGD}{RD} \right)$$

where, GR is the germination rate, NGD is the number of germinated seeds per day, and RD is the related day.

### *2.6. Seedling Variables*

To standardize the seedling sampling, 10 to 15 seedlings that germinated in the first 10 days after the first plant emergence were taken from each genotype. Each seedling was considered as replication. The following parameters were measured after 60 DAS.

### 2.6.1. Seedling Height, Root Length (cm), Number of Leaves (units)

Seedling height (from the stem base to the apex) and root length were measured using a caliper (Mitutoyo, CD-15C, Kanagawa, Japan). Number of leaves were counted for each genotype.

### 2.6.2. Fresh and Dry Biomass (mg)

These parameters were measured using a semi-micro analytical balance (A&D Company, GH-252, Tokyo, Japan). Fresh biomass of aerial (stem + leaves) and root were recorded by separating these two parts of the seedling. To obtain the dry biomass of aerial (stem + leaves) and root, samples were placed in an oven at 80 °C (Sanyo, MOV-112F, Osaka, Japan) for 72 h and then the parameters were recorded.

### 2.6.3. Ratios

Top/root ratio was calculated by the relationship between the seedling height and the root length, whereas top/root biomass ratio was calculated by the relationship between the aerial and root dry weight.

### 2.6.4. Seedling Vigor Indexes

Seedling vigor index I and II were estimated by the following formulas [13]:

$$\text{SVI I} = \text{G} \times \text{MSL}$$

where, SVI I is the seedling vigor index I, G is the germination percentage, and MSL is the mean seedling length.

$$\text{SVI II} = \text{G} \times \text{MSDM}$$

where, SVI II is the seedling vigor index II, G is the germination percentage, and MSDM is the mean seedling (aerial + root) dry mass (expressed in g).

### 2.7. Statistical Analysis

A randomized completed design with a factorial arrangement of 5 passion fruit genotypes × 5 priming methods (including control) was used for the experiment. The experimental unit was constituted by one plastic tray or one seedling of each genotype, depending on the recorded variables.

Data analysis was carried out in the R statistical software version 4.0.4. Shapiro test was used to determine the normality of the data and Levene test was used to assess the equality of variances. Arcsine transformation was applied to emergency (20 days) and germination percentage (30 and 60 days); whereas square root transformation was applied to seedling height, root length, aerial and root weight (both fresh and dry).

A one-way ANOVA function of R, which allows heteroscedasticity of variances, was used to carried out the analysis of the morphological seed variables. While a two-way ANOVA for homoscedasticity of errors was carried out to determine statistical differences among treatments in the factorial arrangement (genotypes × priming methods) for most of the germination and seedling variables. However, due to the variables number of leaves, top/root and biomass ratio did not show homoscedasticity and normality of the residual experimental errors of the model. The Kruskal–Wallis test was used to find statistical differences but using the difference among means for treatments instead of the factors. In addition, the Tukey test at 5% was used to find differences between means; this test was applied for the transformed data in the respective variables as mentioned above. Data showed in the tables refer to the original data for better understanding.

Pearson correlation coefficients between seed and seedling traits were calculated to identify the significant relationships among them.

## 3. Results

### 3.1. Seed Variables

Table 1 shows the characteristics of the seeds of the 5 genotypes assessed. P10 showed the highest seed length (6.79 mm), width (4.54 mm), thickness (1.99 mm), and weight (256.52 mg), followed by INIAP 2009, while PICH1 showed lowest values in these traits.

**Table 1.** Seed traits for each genotype of passion fruit.

| Genotypes | Seed Length (mm) | Seed Width (mm) | Seed Thickness (mm) | Seed Weight (10 Seeds mg$^{-1}$) |
|---|---|---|---|---|
| INIAP 2009 | 6.57 ± 0.05 b | 4.50 ± 0.04 a | 1.95 ± 0.02 a | 255.58 ± 4.52 a |
| P10 | 6.79 ± 0.06 a | 4.54 ± 0.04 a | 1.99 ± 0.03 a | 256.52 ± 5.23 a |
| Gulupa | 5.54 ± 0.02 c | 3.74 ± 0.02 c | 1.81 ± 0.01 b | 186.48 ± 1.41 c |
| POR1 | 6.60 ± 0.03 b | 4.15 ± 0.02 b | 1.97 ± 0.01 a | 226.56 ± 1.53 b |
| PICH1 | 5.22 ± 0.03 d | 3.76 ± 0.03 c | 1.58 ± 0.01 c | 133.22 ± 0.66 d |

Different letters indicate significant differences ($p < 0.05$) using the ANOVA one-way analysis followed by Tukey's test.

### 3.2. Seed Imbibition and Moisture

Seed imbibition and seed moisture varied for each genotype at 0, 12, 24, 48 and 72 h (Figures S3 and S4). INIAP 2009, P10 and POR1 showed a significant increase after 12 h (up to 20%) and after that the incremnet was less substancial, ranging from 10 to 24% in most of the treatments at 48 h. Gulupa and PCH1 showed the sharp increase at 24 h, reaching up to 15% and 22% when they soaked in potassium nitrate.

PEG 6000 showed less imbibition percentage than the other priming methods in all genotypes in most of the evaluated hours; however, it was not statistically different from the other methods in most of the cases (Figure S3). The highest percentages depending on the genotypes after 72 h varied from 20 to 33%. At this time, PICH1 seeds treated with potassium nitrate showed the highest value (32.86%) while POR1 seeds treated with PEG 6000 showed the lowest (11.25%). PEG 6000 usually showed the lowest values for all genotypes at 72 h, ranging from 11 to 16%, except for PICH1, where it reached 22%.

In terms of seed moisture, there were no statistical differences among the priming methods (including imbibition in water) for all the genotypes (Figure S4). The seed moisture varied depending on the genotype and the different priming methods. It started at a range from 6 to 10% depending on the genotype, increased substantially after 24 h, and after that, the increment was less significant at 48 h. After 72 h, the highest percentages ranged from 13 to 18%; at this time, P10 seeds treated with gibberellic acid showed the highest value (18.48%) whereas POR1 seeds treated with PEG 6000 showed the lowest (10.24%). In addition, PEG 6000 also showed the lowest seed humidity for P10 and PICH1 after 72 h of evaluation.

### 3.3. Germination Variables

An additive effect of the factors was observed in the interaction (P × M) for the germination variables (Table 2). The highest emergence percentage at 20 days was reached by P10 seeds treated with PEG 6000 (68%), followed by POR1 treated with hydrogen peroxide and PEG 6000 (47% in both cases). This trend was also observed after 30 days when P10 treated with PEG 6000 reached a high germination (94%) in comparison with the other treatments for all genotypes, followed by POR1 treated with hydrogen peroxide, which showed 89% of germination.

PEG 6000 showed the highest percentages of germination for P10 (97%), Gulupa (82%), and PICH1 (76%), gibberellic acid (90%) and water (92%) for INIAP 2009, and hydrogen peroxide (97%) for POR1; it being P10 and POR1 with PEG 6000 and peroxide of hydrogen that reached the highest germination percentages at 60 DAS. PEG 6000 reached a fast first emergence (12 days) for the genotypes INIAP 2009 and P10. The latter genotype seeds treated with this priming method showed the highest values for all the germination variables. In addition, seeds of genotypes P10 and POR1 treated with potassium nitrate showed acceptable percentages of germination (both 92%).

**Table 2.** Interaction genotype (P) × priming method (M) for the germination variables.

| Treatments | Emergence 20 Days (%) | Germination 30 Days (%) | Germination 60 Days (%) | Day of the First Plant Emergence | Gt 50 (Days) | Germination Rate |
|---|---|---|---|---|---|---|
| P1M1 | 13.89 ± 5.56 bcd | 68.05 ± 2.78 c | 91.67 ± 4.81 ab | 16.00 ± 1.53 abc | 24.00 ± 0.58 abc | 0.86 ± 0.02 c |
| P2M1 | 13.89 ± 5.50 cd | 66.67 ± 2.40 c | 87.50 ± 4.17 abc | 18.67 ± 2.33 abcd | 25.37 ± 1.64 abcd | 0.83 ± 0.09 c |
| P3M1 | 0.00 ± 0.00 d | 13.89 ± 1.39 de | 34.72 ± 2.78 e | 23.00 ± 0.58 cd | 31.17 ± 0.33 bcde | 0.27 ± 0.02 e |
| P4M1 | 19.44 ± 11.11 bcd | 79.17 ± 4.17 abc | 88.89 ± 5.01 abc | 16.00 ± 0.58 abc | 21.96 ± 0.29 ab | 0.94 ± 0.05 bc |
| P5M1 | 0.00 ± 0.00 d | 0.00 ± 0.00 f | 52.78 ± 7.73 cde | 36.00 ± 2.52 f | 48.31 ± 0.69 i | 0.27 ± 0.04 e |
| P1M2 | 2.78 ± 1.39 cd | 20.84 ± 4.17 de | 76.39 ± 1.39 abcde | 18.33 ± 2.19 abcd | 37.94 ± 2.47 efgh | 0.56 ± 0.01 d |
| P2M2 | 1.39 ± 1.09 d | 12.50 ± 6.36 e | 84.72 ± 7.73 abc | 21.33 ± 1.45 bcd | 41.00 ± 0.87 fghi | 0.54 ± 0.07 d |
| P3M2 | 0.00 ± 0.00 d | 11.11 ± 2.78 e | 43.05 ± 2.78 de | 25.67 ± 2.33 de | 39.94 ± 4.91 efghi | 0.27 ± 0.03 e |
| P4M2 | 47.22 ± 3.68 ab | 88.89 ± 3.67 ab | 97.22 ± 1.39 a | 15.33 ± 0.67 abc | 20.06 ± 0.42 a | 1.13 ± 0.02 ab |
| P5M2 | 0.00 ± 0.00 d | 0.00 ± 0.00 f | 0.00 ± 0.00 f | ——* | ——* | ——* |
| P1M3 | 1.39 ± 1.09 d | 33.33 ± 8.33 d | 72.22 ± 7.35 abcde | 21.33 ± 0.88 bcd | 36.00 ± 4.37 efg | 0.54 ± 0.07 d |
| P2M3 | 5.56 ± 4.56 cd | 68.05 ± 2.78 c | 91.67 ± 0.01 ab | 18.33 ± 2.67 abcd | 24.53 ± 0.66 abc | 0.85 ± 0.04 c |
| P3M3 | 2.78 ± 2.08 d | 30.56 ± 1.39 d | 63.89 ± 3.67 bcde | 21.00 ± 2.00 bcd | 31.56 ± 1.11 cdef | 0.50 ± 0.03 de |
| P4M3 | 30.56 ± 7.35 abc | 81.95 ± 2.78 abc | 91.66 ± 4.17 ab | 16.00 ± 1.00 abc | 21.69 ± 0.85 ab | 1.00 ± 0.04 bc |
| P5M3 | 0.00 ± 0.00 d | 0.00 ± 0.00 f | 70.84 ± 4.17 abcde | 35.33 ± 1.20 f | 43.95 ± 2.63 ghi | 0.39 ± 0.04 de |
| P1M4 | 29.17 ± 0.01 abc | 65.28 ± 2.78 c | 81.94 ± 6.06 abcd | 12.33 ± 0.67 a | 22.08 ± 0.94 abc | 0.90 ± 0.03 bc |
| P2M4 | 68.06 ± 5.56 a | 94.44 ± 1.39 a | 97.22 ± 1.39 a | 12.00 ± 0.58 a | 16.49 ± 0.90 a | 1.37 ± 0.01 a |
| P3M4 | 18.06 ± 1.39 bcd | 73.61 ± 1.39 bc | 81.95 ± 2.78 abcd | 18.67 ± 0.33 abcd | 21.92 ± 0.30 ab | 0.85 ± 0.03 c |
| P4M4 | 47.22 ± 3.68 ab | 1.39 ± 1.09 bc | 87.50 ± 4.81 abc | 14.33 ± 0.33 ab | 20.01 ± 0.81 a | 1.00 ± 0.02 bc |
| P5M4 | 0.00 ± 0.00 d | 75.00 ± 0.01 bc | 76.39 ± 11.37 abcd | 32.22 ± 1.76 ef | 43.06 ± 2.17 ghi | 0.43 ± 0.06 de |
| P1M5 | 11.11 ± 9.11 cd | 79.17 ± 4.17 abc | 90.28 ± 2.78 ab | 19.00 ± 1.53 abcd | 25.16 ± 0.86 abc | 0.87 ± 0.07 c |
| P2M5 | 20.83 ± 9.62 bcd | 83.33 ± 2.40 abc | 90.28 ± 5.01 ab | 16.67 ± 0.33 abc | 23.33 ± 0.42 abc | 0.94 ± 0.06 bc |
| P3M5 | 0.00 ± 0.00 d | 19.45 ± 5.01 de | 48.61 ± 8.45 de | 25.33 ± 0.33 de | 34.83 ± 0.44 defg | 0.33 ± 0.05 de |
| P4M5 | 18.05 ± 2.78 bcd | 76.39 ± 3.67 bc | 90.28 ± 6.05 ab | 18.67 ± 0.67 abcd | 23.97 ± 0.22 abc | 0.88 ± 0.05 c |
| P5M5 | 0.00 ± 0.00 d | 0.00 ± 0.00 f | 72.22 ± 3.68 abcde | 38.00 ± 1.15 f | 45.97 ± 1.39 hi | 0.38 ± 0.01 de |

P1 = INIAP 2009, P2 = P10, P3 = Gulupa, P4 = POR1, P5 = PICH1. M1 = water, M2 = hydrogen peroxide, M3 = potassium nitrate, M4 = PEG 6000, M5 = gibberellic acid. Gt 50 = 50% of final germination percentage. * There was no germination. Different letters indicate significant differences ($p < 0.05$) using the ANOVA two-way analysis followed by Tukey's test.

Although the seeds of genotypes P10 (87%) and POR1 (89%) imbibed in water showed less germination percentage (60 DAS) than the seeds treated with PEG 6000, potassium nitrate and hydrogen peroxide, they were not statistically different. On the other hand, Gulupa and PICH1 treated with this priming method showed statistical differences in the percentage of germination (81.95 and 76.39%, respectively) in comparison to the control (34.72 and 52.78%, respectively). In the case of PICH1, it was observed that all priming methods were superior to the control (52.78%), except to hydrogen peroxide, which apparently inhibited the germination. A similar trend was observed for Gulupa, where all the priming methods achieved higher percentages of germination than the control (34.72%).

P10 seeds treated with PEG 6000 and POR1 seeds treated with hydrogen peroxide reached the Gt50 in less time (16 and 20 days, respectively) and also had the highest germination rate (1.37 and 1.13, respectively), while PICH1 showed the longest time (48 days) for the Gt50. Gulupa seeds imbibed in water and treated with hydrogen peroxide, as well as PICH1 seeds imbibed in water showed the lowest germination rate (0.27 in all cases).

### 3.4. Seedling Variables

Seedlings of *P. edulis* f. *flavicarpa* grew bigger than those of *P. edulis* f. *edulis* and *P. maliformis* (Figure S5). INIAP 2009, P10, and POR1 seedlings (yellow passion fruit) showed similar results, which differed from those achieved for Gulupa (purple passion fruit) and the local germplasm PICH1.

In the interaction (Table 3a,b), it was observed that most of the seedlings coming from the seeds treated with PEG 6000 showed high values for seedling height, top and root fresh and dry weight, number of leaves, and seedling vigor index II. In addition, seed imbibed in water of INIAP 2009 and POR1 showed high root length, top fresh weight, top/root dry ratio, and seedling vigor index I, sharing the significance range with other priming methods that showed the highest values. INIAP 2009 and P10 showed high root length (8.97 and 10.02 cm) when they were treated with potassium nitrate. The highest seedling

vigor index I was reached by POR1 treated with hydrogen peroxide (556.85) while the highest vigor index II was reached by P10 treated with PEG 6000 (13.02).

**Table 3.** (**a**) Interaction genotype (P) × priming method (M) for the seedling variables. (**b**) Interaction genotype (P) × priming method (M) for the ratios, indexes, and number of leaves.

| (a) | | | | | |
|---|---|---|---|---|---|
| | **Seedling Height (cm)** | **Root Length (cm)** | **Top Fresh Weight (mg)** | **Root Fresh Weight (mg)** | **Top Dry Weight (mg)** | **Root Dry Weight (mg)** |
| P1M1 | 5.18 ± 0.07 cd | 8.36 ± 0.14 abcde | 351.45 ± 15.03 abcd | 163.80 ± 13.37 cdefg | 76.58 ± 2.77 bcde | 22.89 ± 1.12 defg |
| P2M1 | 4.93 ± 0.10 d | 8.11 ± 0.21 bcde | 343.44 ± 14.93 abcde | 193.01 ± 18.92 bcdefg | 77.60 ± 2.77 bcde | 23.81 ± 2.03 defg |
| P3M1 | 3.27 ± 0.09 gh | 7.82 ± 0.17 bcde | 238.07 ± 17.15 f | 151.76 ± 12.66 efgh | 48.58 ± 3.93 f | 14.81 ± 0.87 hi |
| P4M1 | 5.79 ± 0.11 ab | 9.40 ± 0.36 abc | 363.86 ± 13.48 abc | 122.57 ± 14.56 ghi | 93.23 ± 3.96 abc | 23.63 ± 1.38 defg |
| P5M1 | 2.78 ± 0.12 hi | 5.32 ± 0.36 gh | 57.46 ± 5.70 h | 39.91 ± 10.02 jk | 8.54 ± 1.34 h | 3.43 ± 0.57 k |
| P1M2 | 4.01 ± 0.06 ef | 9.44 ± 0.65 abcd | 314.35 ± 19.49 bcdef | 187.18 ± 15.37 bcdefg | 76.12 ± 4.76 bcde | 23.37 ± 1.74 defg |
| P2M2 | 4.32 ± 0.13 e | 9.97 ± 0.58 ab | 262.76 ± 26.74 ef | 169.69 ± 21.38 cdefg | 58.10 ± 7.91 ef | 17.77 ± 1.67 fgh |
| P3M2 | 2.72 ± 0.10 i | 7.12 ± 0.44 ef | 126.02 ± 15.32 g | 62.36 ± 7.91 ijk | 24.38 ± 3.12 g | 9.39 ± 0.86 ij |
| P4M2 | 5.73 ± 0.13 abc | 9.51 ± 0.31 abc | 386.97 ± 13.26 abc | 242.07 ± 16.79 abcd | 82.97 ± 2.79 abcd | 24.61 ± 1.08 cdef |
| P5M2 | —–* | —–* | —–* | —–* | —–* | —–* |
| P1M3 | 5.13 ± 0.11 cd | 10.02 ± 0.26 a | 384.56 ± 22.19 abc | 256.81 ± 28.18 abc | 87.01 ± 4.50 abcd | 26.85 ± 2.10 abcde |
| P2M3 | 5.22 ± 0.11 bcd | 8.97 ± 0.28 abcd | 384.91 ± 14.70 abc | 228.91 ± 24.25 abcde | 85.85 ± 3.19 abcd | 25.69 ± 1.84 bcdef |
| P3M3 | 4.36 ± 0.23 e | 7.93 ± 0.43 bcde | 307.47 ± 30.46 cdef | 159.57 ± 20.01 defg | 67.75 ± 8.45 def | 19.12 ± 2.15 efgh |
| P4M3 | 5.84 ± 0.07 a | 7.56 ± 0.24 de | 358.35 ± 13.66 abcd | 132.09 ± 14.04 fgh | 93.23 ± 3.02 abc | 22.67 ± 1.82 defg |
| P5M3 | 3.02 ± 0.11 hi | 5.23 ± 0.42 gh | 71.05 ± 5.58 h | 27.80 ± 4.70 k | 11.64 ± 1.04 h | 4.07 ± 0.49 k |
| P1M4 | 5.33 ± 0.09 abcd | 9.46 ± 0.39 abc | 417.00 ± 18.09 a | 276.35 ± 18.00 ab | 102.98 ± 5.89 a | 35.16 ± 1.93 a |
| P2M4 | 5.60 ± 0.08 abc | 9.19 ± 0.34 de | 422.67 ± 20.88 a | 303.82 ± 26.37 a | 99.68 ± 5.77 ab | 34.23 ± 1.94 ab |
| P3M4 | 3.66 ± 0.08 fg | 7.83 ± 0.26 cde | 281.68 ± 12.33 def | 184.05 ± 10.00 bcdefg | 52.37 ± 3.06 f | 16.73 ± 0.85 gh |
| P4M4 | 5.87 ± 0.08 a | 8.30 ± 0.25 abcde | 395.52 ± 16.05 ab | 224.02 ± 20.99 abcde | 94.66 ± 5.44 abc | 32.78 ± 1.59 abc |
| P5M4 | 3.31 ± 0.05 gh | 6.76 ± 0.46 efg | 88.68 ± 8.58 gh | 61.48 ± 11.19 ijk | 15.47 ± 1.79 gh | 5.78 ± 0.80 jk |
| P1M5 | 5.59 ± 0.13 abc | 7.83 ± 0.24 cde | 345.13 ± 12.87 abcde | 151.08 ± 9.21 efg | 79.56 ± 3.62 abcde | 23.49 ± 1.59 defg |
| P2M5 | 5.31 ± 0.05 abcd | 7.51 ± 0.26 de | 342.45 ± 13.31 abcde | 244.00 ± 14.58 abcd | 74.97 ± 2.77 cde | 27.67 ± 1.36 abcd |
| P3M5 | 3.02 ± 0.12 hi | 5.92 ± 0.24 fgh | 141.43 ± 12.47 g | 84.09 ± 9.27 hij | 24.36 ± 2.06 g | 9.39 ± 0.90 ij |
| P4M5 | 5.70 ± 0.12 abc | 8.32 ± 0.32 abcde | 381.48 ± 14.80 abc | 197.04 ± 13.81 bcdef | 92.96 ± 4.48 abc | 22.66 ± 1.74 defg |
| P5M5 | 3.14 ± 0.09 hi | 5.10 ± 0.31 h | 69.73 ± 4.51 h | 38.53 ± 4.27 jk | 12.62 ± 1.02 h | 4.87 ± 0.54 k |

| (b) | | | | |
|---|---|---|---|---|
| | **Top/Root Ratio** | **Top/Root Dry Biomass Ratio** | **Number of Leaves** | **Seedling Vigor Index I** | **Seedling Vigor Index II** |
| P1M1 | 0.62 ± 0.02 cde | 3.58 ± 0.20 abc | 4.14 ± 0.10 fgh | 475.22 ± 24.95 abc | 9.04 ± 0.47 bcde |
| P2M1 | 0.62 ± 0.02 ef | 3.51 ± 0.36 cde | 4.20 ± 0.11 efgh | 431.55 ± 20.55 bc | 8.87 ± 0.42 cde |
| P3M1 | 0.42 ± 0.02 l | 3.26 ± 0.16 cde | 4.58 ± 0.15 bcd | 113.59 ± 9.08 g | 2.20 ± 0.18 h |
| P4M1 | 0.63 ± 0.03 de | 4.08 ± 0.22 ab | 4.73 ± 0.12 abc | 514.51 ± 28.98 ab | 10.39 ± 0.59 bcd |
| P5M1 | 0.55 ± 0.05 ghij | 2.55 ± 0.18 g | 3.40 ± 0.16 i | 146.94 ± 21.53 fg | 0.63 ± 0.09 h |
| P1M2 | 0.42 ± 0.03 kl | 3.31 ± 0.17 cde | 4.09 ± 0.09 gh | 306.35 ± 5.57 de | 7.60 ± 0.14 efg |
| P2M2 | 0.44 ± 0.02 kl | 3.02 ± 0.23 cdef | 3.89 ± 0.20 hi | 365.54 ± 33.37 cd | 6.43 ± 0.59 fg |
| P3M2 | 0.39 ± 0.03 l | 2.57 ± 0.18 g | 4.09 ± 0.09 gh | 117.16 ± 7.55 g | 1.45 ± 0.09 h |
| P4M2 | 0.61 ± 0.03 ef | 3.45 ± 0.18 bcd | 4.73 ± 0.12 abc | 556.85 ± 7.96 a | 10.46 ± 0.15 bcd |
| P5M2 | —–* | —–* | —–* | —–* | —–* |
| P1M3 | 0.52 ± 0.02 ijk | 3.51 ± 0.36 cde | 4.33 ± 0.16 defgh | 370.19 ± 37.67 cd | 8.22 ± 0.84 def |
| P2M3 | 0.59 ± 0.03 efgh | 3.36 ± 0.20 abc | 4.47 ± 0.13 bcdef | 478.86 ± 5.00 abc | 10.12 ± 0.06 bcd |
| P3M3 | 0.56 ± 0.03 fghi | 3.66 ± 0.26 sbc | 4.50 ± 0.19 bcde | 278.76 ± 16.03 de | 5.55 ± 0.32 g |
| P4M3 | 0.78 ± 0.03 a | 4.54 ± 0.41 ab | 4.87 ± 0.17 ab | 535.51 ± 24.34 ab | 10.62 ± 0.48 bc |
| P5M3 | 0.62 ± 0.04 efg | 3.08 ± 0.27 defg | 4.33 ± 0.16 defgh | 214.19 ± 12.60 efg | 1.11 ± 0.07 h |
| P1M4 | 0.58 ± 0.02 efghi | 2.95 ± 0.12 efg | 4.60 ± 0.13 bcd | 436.80 ± 32.28 abc | 11.32 ± 0.84 ab |
| P2M4 | 0.63 ± 0.02 bcde | 2.98 ± 0.17 defg | 4.47 ± 0.17 cdefg | 544.69 ± 7.79 ab | 13.02 ± 0.19 a |
| P3M4 | 0.48 ± 0.02 jkl | 3.15 ± 0.11 cdef | 5.13 ± 0.09 a | 299.60 ± 10.15 de | 5.66 ± 0.19 g |
| P4M4 | 0.72 ± 0.03 ab | 2.94 ± 0.18 efg | 5.00 ± 0.01 a | 513.81 ± 28.24 ab | 11.15 ± 0.61 abc |
| P5M4 | 0.52 ± 0.05 hijk | 2.88 ± 0.36 fg | 4.60 ± 0.31 bcde | 252.81 ± 37.62 def | 1.62 ± 0.24 h |
| P1M5 | 0.72 ± 0.03 ab | 3.59 ± 0.17 abc | 4.40 ± 0.16 defgh | 504.23 ± 15.51 ab | 9.24 ± 0.28 bcde |
| P2M5 | 0.70 ± 0.02 abc | 2.77 ± 0.13 fg | 4.33 ± 0.13 defgh | 479.65 ± 26.61 abc | 9.27 ± 0.51 bcde |
| P3M5 | 0.52 ± 0.03 ijk | 2.85 ± 0.21 efg | 4.13 ± 0.09 fgh | 146.91 ± 25.53 fg | 1.61 ± 0.28 h |
| P4M5 | 0.70 ± 0.03 abcd | 4.40 ± 0.34 a | 4.79 ± 0.11 ab | 227.10 ± 11.55 efg | 10.44 ± 0.70 bcd |
| P5M5 | 0.64 ± 0.04 de | 2.81 ± 0.22 efg | 4.00 ± 0.17 h | 514.30 ± 34.48 ab | 1.26 ± 0.06 h |

(**a**,**b**) P1 = INIAP 2009, P2 = P10, P3 = Gulupa, P4 = POR1, P5 = PICH1. M1 = water, M2 = hydrogen peroxide, M3 = potassium nitrate, M4 = PEG 6000, M5 = gibberellic acid. * There was no germination. Different letters indicate significant differences ($p < 0.05$) using the ANOVA two-way analysis followed by Tukey's test.

In terms of statistical significance compared to water imbibition, P10 seeds treated with PEG 6000 and gibberellic acid (5.60 and 5.31 cm, respectively) and Gulupa seeds treated with potassium nitrate (4.36 cm) showed differences in seedling height in comparison to the control (4.93 cm for P10 and 3.27 cm for Gulupa). Turning to root dry weight (biomass), INIAP 2009, P10, and POR1 treated with PEG 6000 (35.16, 34.23, and 32.78 mg, respectively) showed differences compared to the control (22.89, 23.81, and 23.63 mg, respectively).

### 3.5. Correlation Analysis

There was good correlation between the seed and seedling traits, showing statistical significance in most of the cases (Table S1). Seed length and seed thickness showed significant correlation (>0.90) with all the seedling traits. Seed width showed significant correlation with root fresh weight and root dry weight (0.88 for both). Seed weight showed significant correlation (>0.90) with all seedling traits, except with seedling height. On the other hand, seedling height showed a significant relationship with root length, but it was not significantly correlated with root fresh weight.

## 4. Discussion

### 4.1. Seed Traits

Most commercial plantations of passion fruit are propagated by seed [14]. Passion fruit seed is preserved by small farmers using empirical insights unaware of their morphological and biochemical characteristics and its sensitivity to factors such as mechanical damage, climatic conditions (temperature and humidity), and natural aging, which decrease their quality. Seed quality decreases with the time and rate of deterioration, depending on environmental conditions during storage and the duration that these remain stored; the vigor being the first quality component that is affected, followed by a reduction in the germination or seedling normal growth and finally the death of the seeds [28].

Seed biometry is an adequate strategy both to standardize seedling emergence and to obtain more vigorous seedlings or showing similar size [29]. Morphological descriptors are useful for the characterization of species and this information helps genetic improvement programs [30]. Consequently, it is feasible and necessary to do seed measurements and carry out correlations with characters of agronomic importance [31].

Pasion fruit seeds showed morphological differences according to the genotype but seed traits from the *P. edulis* f. *flavicarpa* (yellow passion fruit) genotypes were similar to each other and differed from the seed of *P. edulis* f. *edulis* and *P. maliformis*, the seeds being larger, thicker, and heavier. In general, seed traits were highly correlated to seedling traits, which is in accordance with the correlation results found by Martins et al. [31]. Seed morphometry is a good strategy to standardize seedling emergency [29]. The identification of relationships between seed and seedling traits is important for obtaining quality seedlings [31].

### 4.2. Seed Imbibition and Moisture

The results of seed imbibition in water of the yellow passion fruit genotypes agree with the trend found by Souto et al. [32]. The last authors used three genotypes of yellow passion fruit and obtained a sharp increase of the seed imbibition until the 12th hour and after that, the increment was less pronounced until the 72nd hour. In this study, the genotypes INIAP 2009, P10, and POR1 (yellow passion fruit) showed the same trend and also there were slight variations in the results within genotypes. On the other hand, Gulupa and PICH1 showed this sharped increment until the 24th hour in some of the treatments; consequently, a response based on the genotype could be inferred.

The initial moisture of the seeds was adequate for its conservation as mentioned by Foschi et al. [26], while seed moisture percentages of the different genotypes varied from 10 to 18% after 72 h, which is less than the usual range of germination (25 and 50% depending on the species) [33], and this might also be related to fact that the aged seeds needed more time to germinate. Seed moisture from fresh seed of passion fruit has showed different values; Cardoma et al. [28] reported 9.5% while Posada [34] obtained 22.4% for *P. edulis*

f. *flavicarpa*, while the latter author reported 22.1% for *P. edulis* f. *edulis* and 38.6% for *P. maliformis*. The aged seed used for this study showed similar seed moisture content to the study of Cardoma et al. [28] for the yellow passion fruit seeds imbibed in water, but for the other species, it was lower. Posada [34] found that, after six months, the seed humidity decreased up to 10.0% for both *P. edulis* f. *flavicarpa* y *P. edulis* f. *edulis*, results that are slightly high to the initial seed moisture found in this research with the 18-month-old seed (from 7 to 10%) imbibed in water.

*4.3. Germination*

Germination and seedling emergence are processes that reactivate metabolic activity of the embryonic axis and cause radicle emergence [32]. Thus, germination is a critical phase that is associated with factors linked to physical, biochemical, and physiological seed processes [35].

Studies about *Passiflora* seeds have reported that they have low and irregular germination [14], and there is a long period between the beginning and end of germination, resulting in unequal seedlings, which becomes a challenge when cultivating the passion fruit species [5]. Low percentages (between 30 and 50%) of germination have been reported for yellow passion fruit [12,13]. Balagera et al. [36] reported low germination (30%) in purple passion fruit and also mentioned that it does not remain its germinative power for a very long period. Therefore, there is differences within species and among genotypes from the same species [37].

Germination of *Passiflora* species can be of genetic origin due to variation between species and genotypes [38]; this response was observed in this research because the different passion fruit species showed different percentages of germination. It is considered that a *Passiflora* genotype has a good percentage of viability when its germination is greater than 80%, due to the fact that they show asynchrony in the germination and may take more than 30 days to germinate, even months [34]. Consequently, the genotypes INIAP 2009, P10, and POR1 (all yellow passion fruits) can be considered the best genotypes because they reached more than 80% of germination with most of the priming methods (including imbibition in water); on the other hand, Gulupa reached that percentage only when treated with PEG 6000.

Light and temperature are critical environmental factors that influence seed germination; however, the mechanisms related to environmental signals are still poorly understood [3]. Pereira and Andrade [39] found that alternating the temperature from 20 to 30 °C improves seed germination compared to constant temperatures; in addition, Souto et al. (2017) [32] also stated that temperatures from 25 to 35 °C contribute to seedling emergence. Thus, the variation of temperature (28 °C ± 3 °C) in the greenhouse might have affected this parameter positively.

Posada et al. [40] reported germination of 87% for *P. edulis* f. *flavicarpa* and 71% for *P edulis* f. *edulis* (both cases fresh seed); the former is similar to the results of this study using aged seeds (between 82 to 91%) for the yellow passion fruit and higher than those for the purple passion fruit (54%).

The seed of *P. edulis* f. *flavicarpa* and *P. edulis* f. *edulis* stored for two years at 4 °C showed average germination of 89 and 57%, respectively [34]. They are similar to those of this research with 18-month aged seeds. In addition, this author [34] mentions that the environmental conditions at the place where the passion fruit (mother plants) was grown influenced its seed germination characteristics, observing variations in the percentages of germination among genotypes grown at different sites. Genotypes INIAP 2009, P10, and POR1, which were grown in the same place, showed relatively similar percentages of germination; these percentages were higher than those of Gulupa and PICH1, which were grown at other sites. Due to that germination was linked to the passion fruit genotypes and also with the origin of the seeds, the right selection of the maternal and paternal parents is an important factor that must be considered in the breeding programs because they will influence the seed quality of the progeny.

Gutierrez et al. [41] have reported high germination percentages (more than 90%) for *P. edulis* f. *edulis* using mechanical scarification. This value is a little higher than that reported in this study using chemical scarification. However, the same authors [41] found low germination percentages (around 19%) for *P. maliformis* using mechanical scarification, which is lower than what was found in this study when seeds were treated with PEG 6000 (82%). It is assumed from these results that passion fruit species that are less domesticated or of a wild type need a pre-germination treatment to improve their germination. This is also corroborated with the fact that the *P. edulis* f. *flavicarpa* genotypes, which are more domesticated, or commercial materials (breeding process) did not show statistical differences in the germination percentages between the seeds imbibed in water and the ones treated by priming methods, which was the opposite for the other two species. Yellow passion fruit genotypes showed the highest percentages of germination, which is in accordance with Lima et al. [42].

Usually, passion fruit seeds start to germinate from 12 to 14 days [34], but the treated aged seed used in this research showed a mean germination time between 24 and 37 days and first day of emergence varied from 16 to 35 days (depending on the genotype), from which it can be inferred that storage time affects the germination speed as is mentioned by Posada [34]. *P. edulis* f. *flavicarpa* seeds have showed germination periods ranging from 15 to 45 days [43]; in this study, INIAP 2009, P10, and POR1 seeds treated with priming methods showed a mean germination time between 24 and 30, and the first day of plant emergence was at around 17 days, thus they were inside the range mentioned above.

In general, *P. edulis* f. *flavicarpa* genotypes reached the Gt50 in less time in comparison to the other passion fruit genotypes assessed in this study. *P. maliformis* was the species that reached the Gt50 in more days (43 to 48 days); this could be related to the fact that this species has a low speed of germination [41]. Although seeds of P10 and POR1 treated with PEG 6000 and hydrogen peroxide, respectively, showed the shortest time to reach the GT50, these results were not statistically different from those obtained with imbibition in water for these genotypes. However, mathematically it represents a differences of nine days for the former genotype, which is a noticeable time; whereas it is just one day for the latter, which would mean that both treatments were more efficient for this genotype.

The highest values for germination index obtained in this study were higher than that (0.99) reported by Cubillos et al. [12] but lower than that (2.02) found by Gurung et al. [13] in passion fruit. *P. edulis* f. *flavicarpa* genotypes showed higher germination index than the genotypes from *P. edulis* f. *edulis* and *P. maliformis*.

### 4.4. Priming Methods

Germination in passion fruit is influenced by the genotypes [37]. In this research, a specific priming method that produces high germination percentages for all the passion fruit genotypes or species was not observed. The genetic variability of genotypes within each species should also be considered [5]; INIAP 2009 and P10 reached the best percentages of germination with different priming methods, even though these two genotypes belong to the same species. Due to this intra- and interspecific variability, aspects about germination percentage and priming treatments need to be discussed in the scientific literature.

Priming is a water-based technique that permits controlled seed rehydration to trigger the metabolic processes generally activated during the early phase of germination [10]. *P. edulis* f. *flavicarpa* seeds imbibed in water for 24 h showed high percentages of germination (around 90%) while *P. edulis* f. *edulis* and *P. maliformis* showed lower percentages. The latter could be associated to the fact that the passion fruit aril contains compounds such as steroids, triterpenoids, and reducing sugars that interfere or reduce water absorption by the seeds, inhibiting seed germination [44]; thus, they need to be treated with a priming method.

Hydrogen peroxide is a major ROS, which plays significant roles in endosperm weakening, mobilization of seed reserves, transmitting environmental cues, and other processes during seed germination [45]. It is used for seed scarification by softening the testa and

stimulating the germination of the viable embryo of the *Passifloras* [18]. Using the same dose and time of immersion, Martínez et al. [18] reported germination of 88% for fresh Gulupa seeds, which is higher than the result of this study with the aged seeds. On the other hand, PICH1 (*P. maliformis*-local germplasm) showed no germination with this priming method; this might be related to hydrogen peroxide acts in signaling in the regulation of seed germination, and the precise regulation of its accumulation by the cell antioxidant machinery is crucial to obtain a balance between oxidative signaling, which encourages germination, and oxidative damage, which prevents or delays germination. Thus, its interactions can be beneficial or deleterious [46]. However, this result was apparently a genotype-response because hydrogen peroxide was an effective priming method for other genotypes.

Nitrogenous compounds such as nitrates encourage seed germination of a wide range of plant species; among them, potassium nitrate is the most widely used chemical for promoting germination [47]. It activates seed germination by creating a water potential equilibrium between seeds and the solution by osmosis; this cation can imbibe water quickly and change the water potential of the seeds [19]. Marostega et al. [25] found that ornamental passion fruit seeds treated with potassium nitrate at 1% for 24 h increased their percentage of germination; in this study, this trend was also observed in all the genotypes, except for INIAP 2009, which showed similar percentage of germination in their seeds treated with potassium nitrate and imbibed in water.

There are almost no studies using PEG (osmotic conditioning) as a priming method for passion fruit seeds. Apoplastic barriers have been speculated to play a crucial role in seed development [48]. As PEG does not enter into the apoplast, water is withdrawn from the cell including cell wall [49]. Osmotic conditioning promotes the accumulation of solutes, resulting in a greater potential for cell turgor during rehydration of the seeds, resulting in the emergence from the primary root in less time [50]. Vieira et al. [8] treated seeds with thermic treatment, then soaked them in PEG 6000 for 36 h and reported similar germination percentages (around 90%) for yellow passion fruit as was found in this research. The last authors [8] also mentioned that soaking seeds in PEG 6000 for more than 36 h decreases the percentage of deterioration and hard seeds, but intervals less than 48 h favors fungus incidence. In general, all passion fruit genotypes treated with PEG 6000 reached percentages of germination around or over 87% when seeds were soaked for 36 h; thus, it can be considered a promising priming method for *Passiflora* species. However, more research is needed to optimize dose and a better understanding of its physiological effects on seeds.

Gurung et al. [13] reported a high percentage of germination (74%) of fresh *P. edulis* seeds treated with 500 ppm of gibberellic acid for 10 min, which was lower than the germination reached for yellow passion fruit but higher than that obtained for the purple passion fruit in this research.

### 4.5. Seedling Development

Passion fruit germination is usually irregular, which influences the growth of seedlings and uniformity, which is also connected with the plant genotype [14]. Water and temperature are essential in seeds germination as well as plant development [51]; thus, these conditions were adequate in this study because all genotypes developed seedlings after the seed emergence.

Seedling growth and vigor are associated to the plant genotype and interact with the environment [52]; therefore, understanding this interaction can enhance passion fruit breeding programs.

Seedling height of Gulupa from seeds imbibed in water was higher than the results obtained by Mabundza et al. [9] from seeds with and without imbibition on water (2.5 and 1.6 cm) at 21 DAF, but there was no difference in terms of number of leaves. The same authors reported higher root biomass at 42 DAS than the results of this study.

Hydrogen peroxide was the priming method that most promoted root length in the different genotypes; this may be related to the fact that it is a signal molecule that mediates

a wide range of physiological and biochemical reactions during the entire period of plant growth, enhancing cell division and promoting root growth [53,54]. The response produced by this compound can be varied depending on the genotype; for example, POR1 seedlings coming from seeds treated with hydrogen peroxide reached the highest value for the seedling vigor index I but Gulupa seedlings treated with the same compound showed the lowest value of this index. INIAP 2009 seedling coming from seeds treated with potassium nitrate showed the greatest root length; this could be associated to the fact that potassium is a fundamental nutrient for cell elongation, thus promoting root growth.

It was observed that PEG 6000 encouraged seedling traits such as seedling height, root length, plant biomass, and number of leaves, which also was reflected in the vigor indexes, which might be related to that PEG promotes stem and root elongation and shoot organogenesis [49,55,56]. Nevertheless, most studies regarding PEG have been carried out in short cycle crops and, thus, more research about the use of PEG in seeds and seedlings of fruit crops is needed to obtain a better understanding of the causes of these beneficial effects and also to optimize dose and times of soaking according to the fruit species.

Gurung et al. [13] reported that *P. edulis* seedlings coming from fresh seeds treated with gibberellic acid (same dose and time as in this study) reached a higher seedling height, number of leaves, fresh and dry shoot weight, vigor index I and II, similar fresh and dry shoot weight, and lower root length than the *P. edulis* genotypes of this study. Due to the results showed above, it can be inferred that seed age might influence the reduction of the seedling vigor. On the other hand, seedlings of yellow and purple passion fruit showed a similar seedling height to those reported by Posada [34] using seeds stored for two years (5.8 cm and 3.5 cm, respectively).

The top/root ratio indicated that the root growth was higher than the passion fruit seedling height, but the top/root dry biomass ratio showed that the top part of the seedling was heavier than the root area.

The results of this research contribute to the knowledge generated to improve the germination and vigor of passion fruit seedlings through the application of priming methods to optimize the use of the seed as a means of propagating this fruit of economic importance.

## 5. Conclusions

In general, PEG 6000 was the priming method that showed high germination percentages for all the passion fruit genotypes assessed in this study, and it apparently also had a positive effect on seedling growth. In terms of genotypes and germination, PEG 6000 was the best priming method for P10 (*P. edulis* f. *flavicarpa*), Gulupa (*P. edulis* f. *edulis*), and PICH1 (*P. maliformis*), while gibberellic acid and water were best for INIAP 2009 (*P. edulis* f. *flavicarpa*) and hydrogen peroxide was best for POR1 (*P. edulis* f. *flavicarpa*), corroborating that the effect of the priming methods on the germination response depends on the genotype. In addition, P10 and POR1 reached acceptable germination percentages when the seeds were treated with potassium nitrate.

Yellow passion fruit genotypes (INIAP 2009, P10, and POR1) obtained adequate germination percentages (around 90%) by imbibing the seed in water for 24 h. Therefore, it constitutes a low-cost efficient method for improving seed germination in this taxon of passion fruit, which is the most commercial in Latin America. On the other hand, it was noticeable that *P. edulis* f. *edulis* and *P. maliformis* seeds needed a priming treatment to enhance their germination.

PEG 6000 positively influenced important seedling traits such as height, root length, and plant biomass, contributing to seedling vigor, especially in the *P. edulis* f. *flavicarpa* taxon (yellow passion fruit), which is the most commercialized in Ecuador.

More research is needed on fruit crops species regarding the utilization of non-common priming methods such as PEG, to know in depth the physiological processes related to the functions of this compound to improve seed germination and producing vigorous seedlings.

**Supplementary Materials:** The following are available online at https://www.mdpi.com/article/10.3390/horticulturae8080754/s1, Table S1: Pearson correlation coefficients between the seed traits and the seedling traits. Figure S1: Seed of five passion fruit genotypes. From left to right (each two seeds): INIAP 2009, P10, POR1 (all *P. edulis* f. *flavicarpa*), Gulupa (*P. edulis* f. *edulis*), and PICH1 (*P. maliformis*). Figure S2: Different stages of seedling germination of *P. edulis* f. *flavicarpa*. The third stage (from the left) represents the emergence of the epicotyl and the fifth stage the developed seedling (germination). Figure S3: Seed imbibition of the five genotypes of passion fruit at 0, 12, 24, 48, and 72 h. M1 = water, M2 = hydrogen peroxide, M3 = potassium nitrate, M4 = PEG 6000, M5 = gibberellic acid. Different letters indicate significant differences ($p < 0.05$) using the ANOVA analysis followed by Tukey's test. Figure S4: Seed moisture of the five genotypes of passion fruit at 0, 12, 24, 48, and 72 h. M1 = water, M2 = hydrogen peroxide, M3 = potassium nitrate, M4 = PEG 6000, M5 = gibberellic acid. Different letters indicate significant differences ($p < 0.05$) using the ANOVA analysis followed by Tukey's test. Figure S5: Seedlings of *P. edulis* f. *flavicarpa* (INIAP 2009, left), *P. edulis* f. *edulis* (Gulupa, center) and *P.1maliformis* (POR1, right). Seeds were treated with PEG 6000.

**Author Contributions:** Conceptualization, W.V. and K.K.; methodology, W.V., T.S., A.S. and K.K.; data curation, W.V. and L.R.; statistical analysis, W.V. and L.R.; writing—original draft preparation, W.V., T.S. and K.K.; writing—review and editing, W.V., T.S., K.K., A.S. and N.T. All authors have read and agreed to the published version of the manuscript.

**Funding:** This research was funded by the Laboratory of Tropical Horticultural Science of Tokyo University of Agriculture. Research grant 46407387H.

**Institutional Review Board Statement:** Not applicable.

**Informed Consent Statement:** Not applicable.

**Data Availability Statement:** Data are contained within the article.

**Acknowledgments:** Thanks to the Japan International Cooperation Agency (JICA) for financing the doctoral studies of the first author through the Program "Agriculture Studies Networks for Food Security (Agri-Net)", to Tokyo University of Agriculture and Instituto Nacional de Investigaciones Agropecuarias (INIAP) for supporting this research. Thanks to Tissa Kannangara for editing this manuscript.

**Conflicts of Interest:** The authors declare no conflict of interest.

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
