# Peer review of "Seed Germination and Seedling Growth of Yellow and Purple Passion Fruit Genotypes Cultivated in Ecuador"

_horticulturae, doi:10.3390/horticulturae8080754_

Round 1
Reviewer 1 Report
General comments:
1) Mention the family to which the species belongs.
2) In general, sentences are very long.
3) There are some repeated sentences (e.g., lines 47-49 and 90-91).
4) The term seminiferous is not commonly used.
5) The objective must be the same in the abstract and the introduction. Also, check its grammatical writing (lines 100-102).
6) Correct repeated words (line 253).
7) Use the same terminology (lines 340-342).
Specific comments
1) Title: I recommend eliminating Long-term conserved seed since this work has not evaluated the effect of seed conservation conditions on germination. For that, they should have also used fresh seeds.
2) What type of seed dormancy do Passiflora species have? The authors mentioned physical dormancy characteristics even though they applied GA, commonly used for physiological dormancy. This work is not about seed dormancy. In addition, the evaluated genotypes did not have dormancy either. I recommend not considering dormancy in this manuscript.
3) Lines 112-113: In what kind of container were the seeds stored (i.e., permeable or impermeable)? It is relevant to estimate the time that viable seeds can remain. The storage for 18 months does not mean that it is long since the seed longevity depends on the species and the storage conditions.
4) Lines 120-122: Fig. 1 is not necessary. I recommend including it as supplementary material. In addition, it needs a graph scale to observe seed size differences.
5) Lines 124-125: clarify why seeds were put in water (I assume to eliminate the empty seeds). Also, indicate that the others were dried and used for the tests. How was the initial viability of the seeds estimated?
6) Line 168: replace the word seed with the seedling.
7) Lines 175-177: Fig. 2 is not necessary. I recommend including it as supplementary material.
8) Line 178: considering that the authors did not discuss Gt50 and GR, I suggest considering only one variable and discussing it. In addition, they considered MGT, which is only in Table 2. It needs revision.
9) Line 191: replace the word plant with the seedling.
10) Lines 215-216: the experimental unit is not the seed but the plastic tray where the seeds were sown.11) Lines 222-223: the heterogeneity of variances can be modeled in one-way or two-way ANOVA.
12) Lines 228-229: the Kruskal-Wallis test requires that the variances be homogeneous.
13) Line 239: The first sentence is not necessary. I suggest deleting it.
14) To be considered in Results: in a factorial experiment, when the factors interact with each other, it is not logical to show level comparisons of each factor separately. What is the meaning of knowing which genotype is better if that response depends on the priming treatment? I suggest rewriting this section.
15) Fig. 3 and 4 are not necessary. Authors only informed the results after 72 h. I recommend including them as supplementary material and reporting some measure of the variability of the data.
16) I recommend including Fig. 5 and Table 4 in the supplementary material.
17) Tables 2, 3a, and 3b have too much information, which makes them difficult to read. I suggest reporting these results as figures, showing genotype x treatment interactions. Also, include statistical aspects in their legend (like Table 1).
18) Mean germination time is not in the Material and Methods section.
19) The discussion section is extensive and describes mechanisms not studied in this work, so I suggest they be eliminated (see details below). Also, the results discussed are unbalanced, e.g., seed germination vs. seedling development.
20) Lines 369-370: these experiments are not appropriate for evaluating seed dormancy break.
21) Lines 373-383: the effect of seed storage conditions on seed germination was not studied in this work.
22) lines 385-390: This is a physiological mechanism not studied in this work. I suggest removing it. I also recommend discussing in more detail the result expressed in lines 391-392.
23) Line 404: What does t/hat mean?
24) Lines 406-415: this paragraph corresponds to seed characteristics more than seed germination.
25) Line 416: replace the word resume with that reactivate.
26) Line 430: replace the word distinct with different.
27) Lines 438-445: seed dormancy was not evaluated in this study.
28) Lines 505-508: This is a physiological mechanism not studied in this work. I suggest removing it.
29) Line 520: Move the sentence “however, seeds….” to the next paragraph, where the PEG effect is discussed.
30) Line 527: Obviously? I suggest changing this word.
31) Line 550: as mentioned before, it is the first mention of MGT. This variable with Gt50 and GR should be discussed in more detail, considering there were no differences between priming and water treatments.32) Lines 555-563: This is a physiological mechanism that has not been studied in this work. I suggest removing it.
33) Lines 568-572: 6°C was not the incubation temperature of seeds (it was the storage temperature), meaning that it did not interact with the GA.
34) Line 576: replace humidity with water.
35) For seedling development, not all the treatments evaluated were discussed.
36) The conclusions should be written again based on suggested comments.
37) References are excessive.
Author Response
General comments
Comment 1: Mention the family to which the species belongs.
Reply: The name of the family has been included in the introduction.
Comment 2: In general, sentences are very long.
Reply: Several sentences has been rewritten.
Comment 3: There are some repeated sentences (e.g., lines 47-49 and 90-91).
Reply: The repeated sentences have been eliminated.
Comment 4: The term seminiferous is not commonly used.
Reply: The term seminiferous has been deleted.
Comment 5: The objective must be the same in the abstract and the introduction. Also, check its grammatical writing (lines 100-102).
Reply: The objective has been corrected in the abstract and introduction, and the grammatical writing was corrected.
Comment 6: Correct repeated words (line 253).
Reply: The repeated words have been eliminated.
Comment 7: Use the same terminology (lines 340-342).
Reply: Terminology has been corrected.
Specific comments
Comment 1: Title: I recommend eliminating Long-term conserved seed since this work has not evaluated the effect of seed conservation conditions on germination. For that, they should have also used fresh seeds.
Reply: “Long-term Conserved Seed” has been deleted from the tiltle and also from the text of the whole manuscript according to the argument of the reviewer.
Comment 2: What type of seed dormancy do Passiflora species have? The authors mentioned physical dormancy characteristics even though they applied GA, commonly used for physiological dormancy. This work is not about seed dormancy. In addition, the evaluated genotypes did not have dormancy either. I recommend not considering dormancy in this manuscript.
Reply: We are agree with the reviewer´s comment and all related to dormancy has been removed from the manuscript.
Comment 3: Lines 112-113: In what kind of container were the seeds stored (i.e., permeable or impermeable)? It is relevant to estimate the time that viable seeds can remain. The storage for 18 months does not mean that it is long since the seed longevity depends on the species and the storage conditions.
Reply: The characteristics of the container for seed storage has been added in the methodology. The “long-term conserved” text has been removed from the manuscript as was recommended before.
Comment 4: Lines 120-122: Fig. 1 is not necessary. I recommend including it as supplementary material. In addition, it needs a graph scale to observe seed size differences.
Reply: Figure 1 has placed as supplementary material. A new picture with a scale has been placed.
Comment 5: Lines 124-125: clarify why seeds were put in water (I assume to eliminate the empty seeds). Also, indicate that the others were dried and used for the tests. How was the initial viability of the seeds estimated?
Reply: It was clarified that the objective of putting the seeds in water was to eliminate empty seeds and others were dried and used for the test.
There was not carried out a viability test for the passion fruit seeds.
Comment 6: Line 168: replace the word seed with the seedling.
Reply: Replacement was done.
Comment 7) Lines 175-177: Fig. 2 is not necessary. I recommend including it as supplementary material.
Reply: Figure 2 was included as supplementary material.
Comment 8: Line 178: considering that the authors did not discuss Gt50 and GR, I suggest considering only one variable and discussing it. In addition, they considered MGT, which is only in Table 2. It needs revision.
Reply: Gt50 and GR results have been mentioned in the text of the results section, as well as they were discussed. MGT was deleted from Table 2.
Comment 9: Line 191: replace the word plant with the seedling.
Reply: This replacement was done in the whole document.
Comment 10: Lines 215-216: the experimental unit is not the seed but the plastic tray where the seeds were sown.11) Lines 222-223: the heterogeneity of variances can be modeled in one-way or two-way ANOVA.
Reply: The experimental unit was corrected. It was indicated that the one-way ANOVA function of R which is used for heteroscedasticity of variances for seed variables (one factor); while a two-way ANOVA for homoscedasticity of errors was done for the interaction of the factors.
Comment 11: Lines 228-229: the Kruskal-Wallis test requires that the variances be homogeneous.
Reply: Kruskal-Wallis is a test which has no restriction for data type. It is a non-parametric test which is not limited due to homoscedasticity or normality of the data. It can be carried out with homogeneous or heterogeneous variances.
Comment 12: Line 239: The first sentence is not necessary. I suggest deleting it.
Reply: the sentence was deleted
Comment 13: To be considered in Results: in a factorial experiment, when the factors interact with each other, it is not logical to show level comparisons of each factor separately. What is the meaning of knowing which genotype is better if that response depends on the priming treatment? I suggest rewriting this section.
Reply: The results about the independent factors has been removed, the suggestion of the reviewer is appropriate. The corresponding section of the results was changed.
Comment 14: Fig. 3 and 4 are not necessary. Authors only informed the results after 72 h. I recommend including them as supplementary material and reporting some measure of the variability of the data.
Reply: Figure 3 and 4 has been added as supplementary files. The variability of the data has been mentioned in the results.
Comment 15: I recommend including Fig. 5 and Table 4 in the supplementary material.
Reply: Figure 5 and Table 4 has been added as supplementary material.
Comment 16: Tables 2, 3a, and 3b have too much information, which makes them difficult to read. I suggest reporting these results as figures, showing genotype x treatment interactions. Also, include statistical aspects in their legend (like Table 1).
Reply: The results of the independent factors has been removed from Tables 2, 3a and 3b and just the results about the interaction has been remained (as reviewer suggested above). We consider that currently the tables are understandable because data was decreased.
Comment 17: Mean germination time is not in the Material and Methods section.
Reply: MGT has been removed from Table 2.
Comment 18: The discussion section is extensive and describes mechanisms not studied in this work, so I suggest they be eliminated (see details below). Also, the results discussed are unbalanced, e.g., seed germination vs. seedling development.
Reply: Changes the discussion have been according to the details mentioned by the reviewer. The discussion about seedling development has been increase to balance it.
Comment 19: Lines 369-370: these experiments are not appropriate for evaluating seed dormancy break.
Reply: All text related to dormancy was removed from the manuscript. This part was rewritten.
Comment 20: Lines 373-383: the effect of seed storage conditions on seed germination was not studied in this work.
Reply: The text has been deleted.
Comment 21: lines 385-390: This is a physiological mechanism not studied in this work. I suggest removing it. I also recommend discussing in more detail the result expressed in lines 391-392.
Reply: the text has been removed. And discussion (lines 391-392) has been improved.
Comment 22: Line 404: What does t/hat mean?
Reply: It was corrected.
Comment 23: Lines 406-415: this paragraph corresponds to seed characteristics more than seed germination.
Reply: The paragraph was moved to the seed traits section.
Comment 24: Line 416: replace the word resume with that reactivate.
Reply: Replacement was done.
Comment 25: Line 430: replace the word distinct with different.
Reply: Replacement was done.
Comment 26: Lines 438-445: seed dormancy was not evaluated in this study.
Reply: The aspect related to seed dormancy was removed. Some information remained because is related to seed germination.
Comment 27: Lines 505-508: This is a physiological mechanism not studied in this work. I suggest removing it.
Reply: The text was removed.
Comment 28: Line 520: Move the sentence “however, seeds….” to the next paragraph, where the PEG effect is discussed.
Reply: The sentence was moved.
Comment 29: Line 527: Obviously? I suggest changing this word.
Reply: The word has been replaced by “apparently”.
Comment 30: Line 550: as mentioned before, it is the first mention of MGT. This variable with Gt50 and GR should be discussed in more detail, considering there were no differences between priming and water treatments)
Reply: The text related to the MGT was deleted. The variables Gt50 and GR were discussed.
Comment 31: Lines 555-563: This is a physiological mechanism that has not been studied in this work. I suggest removing it.
Reply: The text was removed.
Comment 32: Lines 568-572: 6°C was not the incubation temperature of seeds (it was the storage temperature), meaning that it did not interact with the GA.
Reply: The text was deleted.
Comment 33: Line 576: replace humidity with water.
Reply: Replacement was done.
Comment 34: For seedling development, not all the treatments evaluated were discussed.
Reply: Discussion has been improves mentioning all treatments.
Comment 35: The conclusions should be written again based on suggested comments.
Reply: Conclusion has been rewritten.
Comment 36: References are excessive.
Reply: The number of references has decreased after doing the reviewer´s corrections.
Reviewer 2 Report
Dear Authors,
the manuscript contains interesting and valuable results that could make commercial passion fruit cultivation more successful. Nevertheless, I found some shortcomings that should be removed to improve the manuscript. Below is a list of detailed comments and suggestions.
Abstract
Page 1, line 16: The Latin name of the genus Passiflora should be added after “Passion fruit”.
Page 1, line 21: “three different species” is incorrect; you included two species (P. edulis and P. maliformis); I suggest to correct this as “three different taxa” or “two species …, including two forms of P. edulis …”. Please, check throughout the text and correct (e.g., also on page 3, line 102).
Page 1, line 24: If you place the Latin name Passiflora in line 16, you should write in line 24 “P. maliformis”.
Page 1, line 25: The abbreviation of PEG 6000 should be explained.
Introduction
Page 2, line 46: I suggest to correct the beginning of this sentence as: “Passion fruit (Passiflora L.) species …”. Moreover, it would be better to add a brief characteristics of the genus Passiflora (i.e., family, number of species, life form, geographical distribution, number of cultivated species).
Page 2, line 68: “the main commercial species” should be corrected as “the main commercial taxa”. Please, check throughout the text that the two forms (P. edulis f. edulis and P. edulis f. flavicarpa) are not mistakenly considered as two different species.
Page 2, lines 68-70: The authors of the names P. edulis f. edulis, P. edulis f. flavicarpa and P. maliformis should be added (you can use Plants of the World Online as a source of nomenclature).
Materials and methods
Page 3, Seed material: Please, provide more details on seed materials explaining the following questions: What was the origin of seeds/genotypes (where the fruits were collected; only POR1 and PICH 1 are mentioned as local germplasms)? Under what conditions the maternal plants from which the seeds for the research were obtained grew? Did the seeds arise in the fruit spontaneously, did humans support the process of pollination and seed development? What was the initial number of fruits of each genotype from which seeds were obtained for the experiment? Were the fruits/seeds collected randomly? Were the seeds sterilized before storage? What about ploidy level of genotypes used in the study?
Results
Page 3, lines 121-122: The caption of Figure 1 should also include Latin names of examined taxa.
Page 7, lines 250-251: It should be corrected as: “Figures 3 and 4 show seed imbibition and seed moisture reached for each genotype at 0, 12, 24, 48 and 72 hours.
Pages 8-9: The captions of Figures 3 and 4 should be corrected as the figures show five not six genotypes, and “at 0, 12, 24, 48 and 72 hours” not “at o, 12, 24 and 72 hours”. The explanation of small letters showed on the graphs is needed in the captions. Moreover, red color rectangle in the legend of the graphs is not the same color as the line for M4 on the graphs.
Page 12, Table 2: Abbreviation of Gt should be explained.
Discussion
Do the studied genotypes differ in the degree of ploidy? If so, it seems interesting to discuss the influence of polyploidy on the structure and germination of seeds.
With reference to a part of the discussion (lines 452-455), it is worth mentioning the conditions of cultivation/growth of the maternal plants from which the seeds were obtained for research, and discuss the results in this context.
Moreover, some minor editing, grammar and spelling mistakes should be corrected:
Page 2, lines 58-59: “ex-situ” should be corrected as “ex situ”.
Page 2, line 74: “Passifloracea” should be corrected as “Passifloraceae”.
Page 2, line 82: “Gibberellic acid” should be corrected as “gibberellic acid”.
Page 3, line 123: the title of subsection 2.3. should be placed on the next page.
Page 4, lines 135, 145, 146 and other pages: It would be better to leave blank lines between subsection titles.
Page 4, line 136:”plactic trays” should be corrected as “plastic trays”.
Page 6, line 212: The title of section 2.7 should not be bolded.
Page 6, line 228: “diferences” should be corrected as “differences”.
Page 7, lines 239-241: This part of instruction for authors should be deleted.
Pages 10-11: A comma is needed before the word “respectively” (e.g. lines 297, 298, 311, 315). Also, on page 4, line 138.
Page 16: A blank line should be added between Table 3 and Table 4.
(Page numbering of Discussion does not follow the order of the rest of the text).
Page 16, Table 4: Word spacing in the upper line of variables is unequal (i.e., from Top fresh weight to Root dry weight).
Page 17, line 397: “Seed moisture form fresh seed…” should be corrected as “Seed moisture from fresh seed…”.
In section 4.3. some paragraphs are incorrectly divided/placed (compare the lines 416 and 420; lines 438 and 446).
I am not sure if the style of some citations in the text is correct (e.g., page 2, line 64: “Doijode (2001) [14]; maybe the year is not needed) – please, check the style of references throughout the whole text.
Author Response
Comment 1: Page 1, line 16: The Latin name of the genus Passiflora should be added after “Passion fruit”.
Reply: “Passiflora” has been added.
Comment 2: Page 1, line 21: “three different species” is incorrect; you included two species (P. edulis and P. maliformis); I suggest to correct this as “three different taxa” or “two species …, including two forms of P. edulis …”. Please, check throughout the text and correct (e.g., also on page 3, line 102).
Reply: This correction has been done in the whole text of the manuscript.
Comment 3: Page 1, line 24: If you place the Latin name Passiflora in line 16, you should write in line 24 “P. maliformis”.
Reply: The correction has been done.
Comment 4: Page 1, line 25: The abbreviation of PEG 6000 should be explained.
Reply: The full name of the abbreviation PEG 6000 has been added.
Comment 5: Page 2, line 46: I suggest to correct the beginning of this sentence as: “Passion fruit (Passiflora L.) species …”. Moreover, it would be better to add a brief characteristics of the genus Passiflora (i.e., family, number of species, life form, geographical distribution, number of cultivated species).
Reply: the correction has been done and the information about the characteristics of the genus Passiflora has been included.
Comment 6: Page 2, line 68: “the main commercial species” should be corrected as “the main commercial taxa”. Please, check throughout the text that the two forms (P. edulis f. edulis and P. edulis f. flavicarpa) are not mistakenly considered as two different species.
Reply: The correction was done and it was checked that the two forms (P. edulis f. edulis and P. edulis f. flavicarpa) were not considered as two different species.
Comment 7: Page 2, lines 68-70: The authors of the names P. edulis f. edulis, P. edulis f. flavicarpa and P. maliformis should be added (you can use Plants of the World Online as a source of nomenclature).
Reply: The names of the authors has been added.
Comment 8: Page 3, Seed material: Please, provide more details on seed materials explaining the following questions: What was the origin of seeds/genotypes (where the fruits were collected; only POR1 and PICH 1 are mentioned as local germplasms)? Under what conditions the maternal plants from which the seeds for the research were obtained grew? Did the seeds arise in the fruit spontaneously, did humans support the process of pollination and seed development? What was the initial number of fruits of each genotype from which seeds were obtained for the experiment? Were the fruits/seeds collected randomly? Were the seeds sterilized before storage? What about ploidy level of genotypes used in the study?
Reply: All the information asked by the reviewer has been added in the section of plant material. Seeds were not sterilized prior conservation. The species and taxa used in this study were all diploid.
Comment 9: Page 3, lines 121-122: The caption of Figure 1 should also include Latin names of examined taxa.
Reply: The names of the taxa were included in the caption of Figure 1; however, this figure was placed as supplementary material for suggestion of another reviewer.
Comment 10: Page 7, lines 250-251: It should be corrected as: “Figures 3 and 4 show seed imbibition and seed moisture reached for each genotype at 0, 12, 24, 48 and 72 hours.
Reply: the correction has been done; however, these figures were placed as supplementary material for suggestion of another reviewer.
Comment 11: Pages 8-9: The captions of Figures 3 and 4 should be corrected as the figures show five not six genotypes, and “at 0, 12, 24, 48 and 72 hours” not “at o, 12, 24 and 72 hours”. The explanation of small letters showed on the graphs is needed in the captions. Moreover, red color rectangle in the legend of the graphs is not the same color as the line for M4 on the graphs.
Reply: The corrections were done according to the reviewer´s suggestions. However, these figures were placed as supplementary material for suggestion of another reviewer.
Comment 12: Page 12, Table 2: Abbreviation of Gt should be explained.
Reply: The abbreviation Gt50 has been explained at the endo of the table.
Comment 13: Do the studied genotypes differ in the degree of ploidy? If so, it seems interesting to discuss the influence of polyploidy on the structure and germination of seeds.
Reply: The accessed genotypes are all diploid, thus there is not difference in the degree of ploidy.
Comment 14: With reference to a part of the discussion (lines 452-455), it is worth mentioning the conditions of cultivation/growth of the maternal plants from which the seeds were obtained for research, and discuss the results in this context.
Reply: The conditions of the cultivation of the maternal plants was mentioned in the section of plant material. The results were discussed.
Comment 15: Page 2, lines 58-59: “ex-situ” should be corrected as “ex situ”.
Reply: The correction was made.
Comment 16: Page 2, line 74: “Passifloracea” should be corrected as “Passifloraceae”.
Reply: The correction was made.
Comment 17: Page 2, line 82: “Gibberellic acid” should be corrected as “gibberellic acid”.
Reply: The correction was made.
Comment 18: Page 3, line 123: the title of subsection 2.3. should be placed on the next page.
Reply: The correction was made.
Comment 19: Page 4, lines 135, 145, 146 and other pages: It would be better to leave blank lines between subsection titles.
Reply: The correction was made.
Comment 20: Page 4, line 136:”plactic trays” should be corrected as “plastic trays”.
Reply: The correction was made.
Comment 21: Page 6, line 212: The title of section 2.7 should not be bolded.
Reply: The correction was made.
Comment 22: Page 6, line 228: “diferences” should be corrected as “differences”.
Reply: The correction was made.
Comment 23: Page 7, lines 239-241: This part of instruction for authors should be deleted.
Reply: The correction was made.
Comment 24: Pages 10-11: A comma is needed before the word “respectively” (e.g. lines 297, 298, 311, 315). Also, on page 4, line 138.
Reply: The correction was made in the whole document.
Comment 25: Page 16: A blank line should be added between Table 3 and Table 4.
Reply: The correction was made.
Comment 26: Page 16, Table 4: Word spacing in the upper line of variables is unequal (i.e., from Top fresh weight to Root dry weight).
Reply: The correction was made.
Comment 27: Page 17, line 397: “Seed moisture form fresh seed…” should be corrected as “Seed moisture from fresh seed…”.
Reply: The correction was made.
Comment 28: In section 4.3. some paragraphs are incorrectly divided/placed (compare the lines 416 and 420; lines 438 and 446).
Reply: Paragraph indentation has been done and it was checked in the whole document.
Comment 29: I am not sure if the style of some citations in the text is correct (e.g., page 2, line 64: “Doijode (2001) [14]; maybe the year is not needed) – please, check the style of references throughout the whole text.
Reply: This style of citation has been checked with other papers published in the journal and the year has been removed.
Round 2
Reviewer 1 Report
The new version of the manuscript is in condition to be published.